# GA-Based Permutation Logic for Grid Integration of Offshore Multi-Source Renewable Parks

Brenda Rojas-Delgado [1,*] , Chisom Ekweoba [1], George Lavidas [2] and Irina Temiz [1]

1 Department of Electrical Engineering, Division of Electricity, Uppsala University, Box 65, 75103 Uppsala, Sweden

2 Faculty of Civil Engineering and Geosciences, Department of Hydraulic Engineering, Offshore Engineering Group, Marine Renewable Energies Lab, Delft University of Technology, Stevinweg 1, 2628 CN Delft, The Netherlands

* Correspondence: brenda.rojas@angstrom.uu.se; Tel.: +46-18-471-5802

**Abstract:** This paper proposes and analyzes a genetic algorithm based permutation control logic applied to the aggregator of an offshore multi-source park. The energy losses at the common coupling point are accounted for in the feedback. This paper focuses on offshore distributed energy resources, such as floating photovoltaic (PV), wind, and wave power. The main contributions of this research are the development of a control system that is capable of tracking the set-point imposed by the demand curve for each source individually, the introduction of a capacity factor for combined offshore floating PV/wind/wave power farms, and the unveiling of pure offshore renewable sources as potential storage-less flexibility service providers. The results of a case study for a site near San Francisco showed that energy losses and capacity factors are positively influenced by implementing the proposed approach.

**Keywords:** distributed energy resources; genetic algorithm; offshore renewables; permutation logic; point of common coupling

## 1. Introduction

According to the World Energy Outlook (WEO) 2021, if a conservatively stated policies scenario (STEPS) is the one considered, electricity demand by 2050 will increase to 42,000 TWh, 80% above 2020's level, whereas the total generation will account for 46,703 TWh, from which wind, solar photovoltaic (PV), and marine energy combined represent two thirds [1]. Moreover, several approaches have been undertaken to manage the rise in variable renewable energies, with particular emphasis on smoothing the fluctuating power before reaching the point of common coupling (PCC) towards either mainland or islanded grids, these approaches being more storage-side- or demand-side-managed than source-side-managed [1–4].

Direct current (DC) link voltage control and converter control approaches can surpass the disadvantages attained to the batteries when dealing with naturally harmonic primary sources, such as wind and wave power. Nonetheless, carrying out such tactics makes it difficult to determine the actual size of either flywheels or capacitors to cope with oscillations, which is the easiest thing to do, assuming a size and performing algorithms with such takeovers [5]. Pitch-based control approaches can capture the maximum amount of energy, especially when the control is exerted individually on each blade, apart from generating revenues on a wide range of their capacity [6–9]. However, the high percentage of failure on such systems can lead to an increment in their maintenance cost, apart from wearing out at a faster pace of the braking systems when used to curtail the exceeding generation [10–12]. Lastly, demand-side management (DSM) approaches, such as peak shaving, demand response, valley-filling, and load-shifting, are considered less costly to smooth demand curves. However, intentional influencing on end-user clients' consumption

patterns, privacy concerns related to the unauthorized treatment of metering data, and a lack of reliable demand forecasts based on real-time stochastic renewable sources allowing to simplify the pricing schemes are accounted as the main disadvantages of DSM strategies and methodologies [13–15].

The ultimate goal of DSM and generation systems management is to match demand and supply. The difference between both strategies is that the former focuses on altering the demand profile to postpone the augmentation of distributed energy resources (DER). The latter tracks the load demand by expanding, relocating, or resizing the generation systems while enlarging the transport systems [1,5,14–17].

Apart from the methods mentioned above, hybrid implementation of renewable energy sources (RES) is deemed a good solution for several problems, such as reduction of resources variability, storage sizing, and increased power supply reliability to cope with the imbalance between wind and solar sources due to geographic constraints [5,16,18].

Henceforth, the need for the development and integration of multi-source parks. The onshore PV/wind hybrid model can be considered the most commonly developed combined power park solution because both sources have already reached grid parity. Such an arrangement poses the advantages of reduced power fluctuation while constituting a realistic solution for electrical generation in islanded areas. Still, its reliability and competitiveness, compared to fossil-fuel power sources, vanishes if no backup controllable generation or storage system is present in the power system [3,16,19].

Concerning the combination of wind and wave parks, potential benefits such as reduced power variability, better predictability, and shared costs have been reported. However, these arrangements depend on a correctly sized storage system for energy autonomy and surplus purposes. The challenges can differ depending on whether the power parks will be separate or combined [20–22]. With regards to combining PV and wave power parks, few hybrid systems of this nature have been found in the literature. The main insight is that seasonal complementarity makes this combination ideal at locations where solar energy is abundant, and waves allow it to capture more power. No research has been found related to the combination of offshore floating PV (OFPV) and wave power parks [20].

The performance of the different hybrid renewable sources has been improved and reinforced with the help of optimization techniques, such as integer linear programming, particle swarm optimization, game theory, genetic algorithm (GA), etc. For DSM and generation sizing/relocation, optimization techniques are based on control strategy, decision variable, pricing scheme, and included uncertainties. However, the main targets when optimizing DSM strategies are minimizing costs and maximizing welfare [15]. Conversely, the objectives can be more diverse for generation systems, including minimization of costs, power curtailment, power losses, and voltage deviation while maximizing energy production [23].

Amongst all the available optimization techniques, GA can handle non-linear oscillations and allow the size of remote-sited integrated energy systems even if the weather data are unavailable [16], which is suitable for pure offshore power parks. GA is a process that imitates the natural selection process. It allows obtaining different solutions for the same problem, which makes it also suitable for control systems that have been designed under the principles of genetic programming [24], apart from being scalable toward multi-objective optimization processes. Implementing GA on multi-source parks avoids forcing each source to work at its highest generation point when load demand is lower than the rated power. These benefits allow GA to overcome its limitations, such as slow convergence, longer execution time, and trend to convergence towards local optima [16–19,25].

This paper proposes a novel GA-based permutation logic for grid integration of offshore multi-source renewable parks, which has been applied to a three-source aggregator designed under the principles of genetic programming. The results of a case study near the San Francisco Bay Area have shown that, under certain hypotheses, energy losses and capacity factors are positively influenced by the total or partial disconnection of

offshore power parks according to the implemented permutation logic. The key finding of this research is that there are other alternatives to reduce energy losses than totally disconnecting renewable generation units when the demand is tracked, relying excessively on storage systems to prevent curtailments, or forcing the generation systems to work at their maximum point, are no longer the best approaches to embark on.

The main contributions of this paper are summarized as follows:

(A)　Energy loss reduction is achieved by implementing the proposed commutation logic, which entails reducing costs associated with grid integration.
(B)　An energy output smoothing method is proposed to control an offshore multi-source park by a unique closed loop, avoiding the need to implement separate smoothing techniques for each power park.
(C)　An islanded generation system consisting of offshore floating photovoltaic, wind, and wave power (OPWW) parks integrated into the mainland grid is considered.
(D)　The potential of offshore renewable sources as electricity flexibility service providers, whose coordinated scheme with the distributed system operator (DSO) at the PCC is non-storage-dependent, is unveiled.
(E)　A combined capacity factor is calculated for each performed permutation and later optimized to reduce seasonal variability.
(F)　A seasonal GA-based permutated control strategy is suggested, where the set-point imposed by the demand curve can be tracked at an individual pace for each source.

## 2. Methodology and Calculations

### 2.1. Studied Area and Power Profiles

#### 2.1.1. Installation Site under Study

The installation site for this research is the US west coast, particularly the San Francisco Bay Area. The location of the multi-source park is assumed to be the same as the site where significant wave height and wave energy period data have been gathered [26].

An approximated representation of the multi-source power park, which initially includes PV, wind, and wave sources, as well as a power transformer connected between the offshore site and the onshore power system, is depicted in Figure 1.

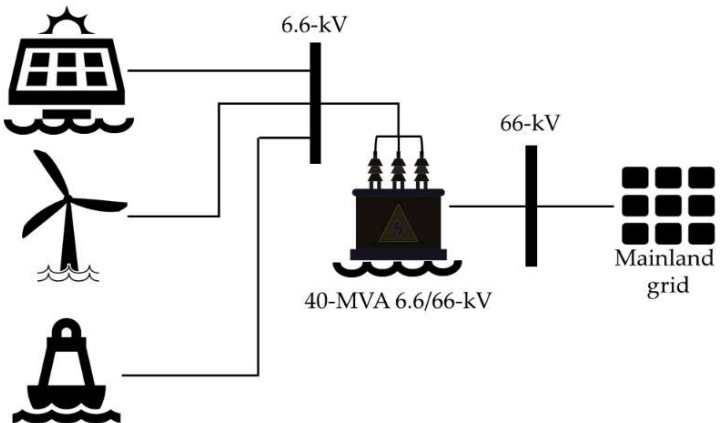

**Figure 1.** Multi-source park studied in the proposed approach.

For the sake of graphic uniformity, each magnitude plotted throughout the paper will be represented with a different color. OFPV energy will be fuchsia, wind energy will be cyan, wave energy will be gold, and so forth.

#### 2.1.2. Generation Profile

The generation energy profile database consists of 3 hourly universal time coordinated (UTC) energy profiles corresponding to OFPV, offshore wind, and wave power systems. The data provided were captured from power plants whose design-rated values are 200 (PV),

3000 (wind), and 600 kW (wave). The dataset was provided by the Marine Renewable Energies Lab of TU Delft (www.tudelft.nl/ceg/mrel) (accessed on 6 December 2022).

The PV panel used in the original dataset is Mitsubishi mono crystalline 250 Wp @ STC: 1000 W/m², 25 °C. The wave power matrix is 600 kW nominal (WaveStar), and the wind turbine is the Vestas 3 MW. The treatment of the whole dataset consists of normalizing the values concerning the installed capacity of each source and applying reverse engineering design criteria to obtain the desired power from the irradiance and wind speed. However, the data related to wave power is only scaled up to the desired rated value.

For this research, the power profiles are individually normalized according to the previously mentioned installed capacity and later scaled up to the desired rated powers. Hence, the rated power of each studied power park is discriminated like 1-MW OFPV, 12-MW offshore wind, and 5-MW wave power. The three generation systems are connected to a 40-MW 6.6/66 kV power transformer whose charge factor is assumed to be 50%, whereas the charge factor of both transmission and distribution cable is taken as 75%.

Figure 2 portrays each power park's energy profiles in kWh after normalization and scaling up to the desired rated power. These profiles are illustrated from 1 January to 31 December 2016.

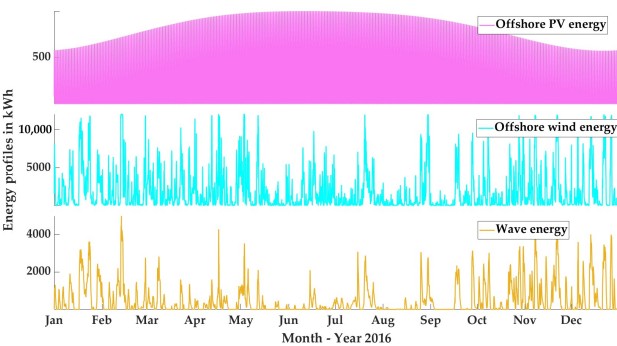

**Figure 2.** 2016 energy profiles of the studied area in kWh.

### 2.1.3. Demand Profile

The demand profile is obtained from [27] and later normalized according to the peak demand registered back in 2016, whose representation is illustrated in Figure 3. Outliers have been suppressed.

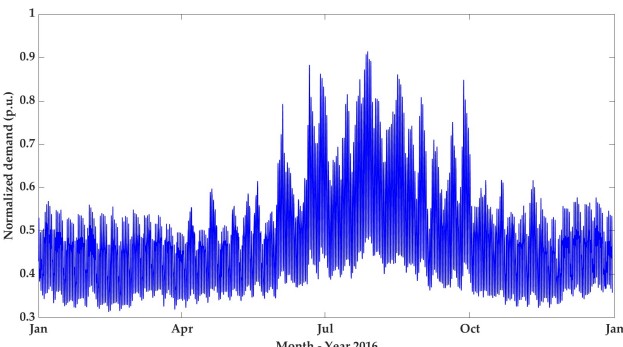

**Figure 3.** 2016 demand profile (normalized).

However, it is treated differently before and after performing the optimization:

(a)  When the permutation logic is performed without optimization, the load at the PCC is assumed to have a Y-connected resistive impedance, whose rated value is 0.6 Ω [28], and varies according to the demand profile, with the current flowing throughout the whole power system. Hence, demand and generation curves are shifted.

(b)  For the optimization process, the normalized demand is multiplied by a factor equal to the average value of the total generation. Thus, both curves are no longer shifted.

### 2.2. Main Components of the Studied Multi-Source Park

2.2.1. OFPV Generation

The PV generation profile is normalized and later escalated for an expected rated power of 1-MW containing AEC mono crystalline glass–glass PV panels AS-M1202Z-BH(M6)-375/HV, @ STC: 1000 W/m$^2$, 25 Celsius. To obtain the desired profile, it is also necessary to calculate the appropriate amount of PV inverters for grid integration purposes.

- PV panels

The PV panels used as a reference for the simulations are the AEG Photovoltaic Module AS-M1202Z-BH (M6 CELLS), whose main numeric features are depicted in Table 1. These PV panels are manufactured with glass–glass technology and a standard junction box IP68, making them suitable for placing at offshore floating power parks since they are protected from high moisture levels.

**Table 1.** Main characteristics of the PV panels used for simulations.

| Parameter | Value |
|---|---|
| Rated power (Wp) | 375 |
| Vmp (V) | 34.10 |
| Imp (A) | 11.01 |
| VOC (V) | 41.89 |
| ISC (A) | 11.43 |
| Module Efficiency (%) | 20.59 |
| Maximum DC voltage (V) | 1500 |

- Power inverter

The inverter used for simulations is the FIMER Solar inverter PVS-350-TL, whose DC input data are given in Table 2. The reason behind choosing this device, in particular, is to comply with technical criteria imposed by the European Scalable Complementary Offshore Renewable Energy Sources (EU-SCORES) project [29].

**Table 2.** Solar inverter's DC input data.

| Parameter | Value |
|---|---|
| Absolute maximum DC input voltage (V) | 1500 |
| Rated DC input voltage (V) | 1080 |
| Number of independent MPPT | 12 |
| Maximum DC input current for each MPPT (A) | 45 |
| Maximum input short circuit current for each MPPT (A) | 60 |
| Number of DC input pairs for each MPPT | 2 |

To calculate the number of inverters needed for a 1-MW OFPV park, it is necessary to estimate how many PV panels can be placed at each DC input.

For doing so, several assumptions are made:

(a) PV panels at each row are connected only in series.

Each DC input carries only series arrangements, whereas the pairs for each maximum power point tracking (MPPT) are connected in parallel.

(b) Calculations are based on the rated DC input voltage instead of the absolute maximum DC voltage.

(c) Vmp and Imp are the voltage and current values taken to calculate the total rated power summed by the panels connected to each inverter. Short-circuit (SC) current and open-circuit (OC) voltage are only used for compliance purposes.

The irradiance data is taken from the US Energy Information Database, and the particular location used as a reference is San Francisco Bay Area [27]. This same assumption is made concerning wind and wave data.

Considering the assumptions made, the data displayed in Tables 1 and 2, and the calculations performed, it is determined that 30 PV panels connected in series forming 24 strings connected in parallel are needed to sum up 270 kW of total power coming from the PV panels to the DC side of each required inverter.

Besides, it is also determined that 3 PV inverters are needed to comply with the main criterion of 1 MW total installed power and with both SC and OC current constraints.

- Connection to the power transformer

Integrating the OFPV park into the grid implies the utilization of direct/alternate current (DC/AC) inverter units, whose selection criteria are the capability of operating under either SC or OC conditions and compliance with cable constraints at the AC side of the inverter. Besides, not only the efficiency of the modules has to be considered but also the efficiency of the inverters. For calculations, the efficiency of the inverters is assumed to be the lowest of the two stated in the inverter's specification sheet. Hence, if assuming that the efficiency of each inverter used is the weighted efficiency, the power output that can be delivered at the AC side, for an assumed total rated power of 1 MW, is calculated with the following equation:

$$P_{\text{PV}_{AC-side}} = P_{DC} \cdot \eta_{EURO} \cdot N_{inv} \tag{1}$$

where $P_{\text{PV}_{AC-side}}$ is the power delivered by the OFPV park at the AC side, $P_{DC}$ is the power delivered by each inverter at the DC link once the PV panels are connected, $\eta_{EURO}$ is the weighted efficiency, and $N_{inv}$ is the number of inverters required. It is important to note that, under either SC or OC conditions, the PV panels do not surpass the rated power of the inverters, which is 350 kW at the unitary power factor.

### 2.2.2. Offshore Wind Turbine

For the wind turbine case, a wind power park consisting of only one wind turbine is considered. The power curve utilized for this research corresponds to the wind energy profile provided by TU Delft and later normalized to be scaled up to 12 MW, the same rated power of a GE General Electric HALIADE-X 12 MW model [30] (See Figure 4).

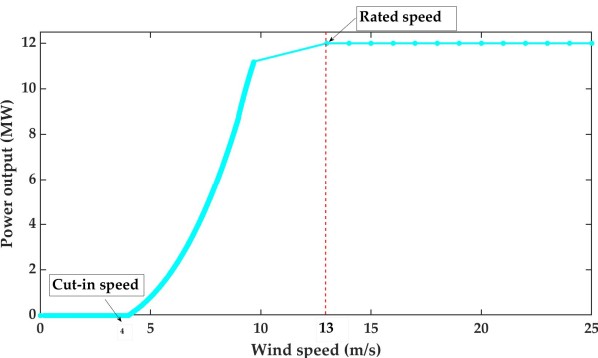

**Figure 4.** Power output curve used for the research.

Thus, the wind speed for the aforementioned model is calculated with:

$$P_{wind} = \left(\frac{1}{2}\right) \cdot Cp \cdot \rho_{air} \cdot A_{swept} \cdot v_{out}^3 \tag{2}$$

where $P_{wind}$ is the power delivered by the wind turbine or the useful wind power after trespassing the swept area, $Cp$ is the power coefficient, which has been assumed equal to the Betz limit (16/27), $\rho_{air}$ is the air density (1.2 kg/m$^3$), $A_{swept}$ is the area swept by the blades (38,000 m$^2$), and $v_{out}$ is the wind speed registered after the swept area.

### 2.2.3. Wave Energy Converters (WECs)

In the case of the wave power park, WaveStar technology is used with the dataset normalized up to the desired rated power (5 MW). However, no new calculations considering the wave energy period and significant wave height are performed because there is no clear indication of the actual depth in which these buoys are moored at the studied location [26]. Thus, it is opted to use the wave energy profile as TU Delft has provided it for validity reasons. Nonetheless, it is essential to point out some takeovers:

(a) The data are taken from the Station 46327 wave measurement buoy—San Francisco Bay, California [26].

(b) The total power of the wave power park is assumed to be 5 MW, with each wave energy converter rated at 30 kW.

(c) No overload conditions are considered, or their effects are taken into account for aggregation purposes [5,23,31].

### 2.2.4. Offshore Transmission System

- Power transformer

The three offshore power parks are connected to a 40-MW 6.6/66 kV power transformer, and both transmission and distribution cable arrangements have rated voltages of 6.6 kV and 66 kV, respectively. The datasets utilized to design the offshore substation were provided by the company WavEC Offshore Renewables under the frame of Project DTOceanPlus [32].

The main substation (after the aggregator) is assumed to be mounted on an offshore platform, with all the cables of subsea type. Mainland connections of the power transformer or any other topologies are out of the scope of this paper. Further, no hydro or aerodynamic factors affecting the performance of the multi-source park are considered.

There are several reasons behind the selection of such transformer units:

(a) The charging factor assumed for the power transformer is 50% of the installed capacity of the three farms sums up to 18 MW; therefore, the combined sources are only needed to occupy up to 45% of the total capacity of the power transformer.

(b) To avoid using more transformers, it is assumed that the downstream voltage is the same as the studied wind turbine (6.6-kV). However, the inverter used for the studies has a rated AC grid voltage ($V_{ac,r}$) of 800-V, so it has to be assumed that the PV panels are connected to a dedicated 1000-KVA 6.6/0.69-kV distribution transformer sharing the power park with the main transformer.

(c) In general, larger losses are registered for lower voltage levels due to a higher current value. However, results drastically change when the losses are studied on different components needed to connect wind turbines to the grid. For example, losses at the WT transformers and sea-to-land cables are directly proportional to voltage, whereas the collection grid losses are the opposite. On top of that, losses at substation transformers vary slightly, as well as the efficiency at nominal power [33]. In such a case, 66 kV offers a perfect balance between losses and efficiency without incurring higher costs due to transport system over-dimensioning.

- Offshore 6.6-kV cable arrangement

Two main criteria are considered for choosing the best cable arrangement at the downstream side of the transformer: current rating and costs. After matching these two criteria with the rated current of the transformer downstream, it is determined that the best cabling option is 25 cables of 145 A each, intending to respect the imposed charging factor (75%).

- Offshore 66-kV cable arrangement

The transmission cable arrangement is subjected to costs and voltage ratio constraints. Hence, the best option is to place one 420-A 66-kV cable between the transformer and the PCC.

*2.3. Methodology Employed*

　　The grid integration of the multi-source offshore park depicted above is assessed in the following steps:

(a)　The three sources are aggregated to the grid with a combination of sources based on a permutation logic, where each offshore power park is connected totally or partially to the power transformer from the aggregator side. This approach also implies total disconnection of one or several sources.

(b)　The aggregation study contemplates the calculation of average and peak power at the PCC. That part will also include the analysis of the peak/average performance ratio and normalized standard deviation [23,31].

(c)　A GA-based permutation logic is implemented.

(d)　The aggregation of the multi-source park with and without implementation of the optimization technique is compared for different seasons.

(e)　Permutated capacity factors are calculated.

(f)　Battery energy storage system (BESS) size is determined after performing the GA search.

　　It is important to point out that, for the sake of all the studies mentioned above, economic aspects related to implementing the optimization technique are out of the scope of this research.

*2.4. Methods*

2.4.1. Permutation Logic Analysis through the Aggregation of Multiple Offshore Renewable Energy Sources

　　The first attempt to prove the efficiency of the proposed approach is to implement the permutation logic without applying any optimization technique. Each source park is connected to an aggregator that has the property of switching them off either totally or partially following a truth table [34]. The permutation logic is based on the principles of genetic programming [24], which consists of "mimicking" the behavior of an electronic multiplexer, namely, the 3-8-line decoder, where gates control three different inputs to produce eight different outputs [35], whose truth table is represented in Figure 5.

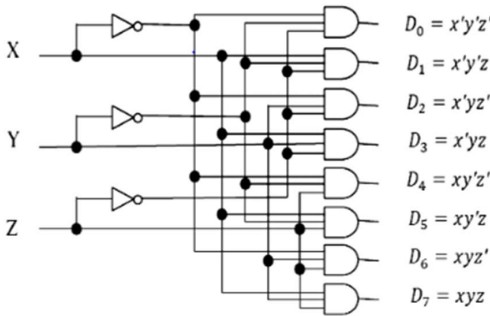

| Input | | | Output | | | | | | | |
|---|---|---|---|---|---|---|---|---|---|---|
| X | Y | Z | D0 | D1 | D2 | D3 | D4 | D5 | D6 | D7 |
| 0 | 0 | 0 | 1 | 0 | 0 | 0 | 0 | 0 | 0 | 0 |
| 0 | 0 | 1 | 0 | 1 | 0 | 0 | 0 | 0 | 0 | 0 |
| 0 | 1 | 0 | 0 | 0 | 1 | 0 | 0 | 0 | 0 | 0 |
| 0 | 1 | 1 | 0 | 0 | 0 | 1 | 0 | 0 | 0 | 0 |
| 1 | 0 | 0 | 0 | 0 | 0 | 0 | 1 | 0 | 0 | 0 |
| 1 | 0 | 1 | 0 | 0 | 0 | 0 | 0 | 1 | 0 | 0 |
| 1 | 1 | 0 | 0 | 0 | 0 | 0 | 0 | 0 | 1 | 0 |
| 1 | 1 | 1 | 0 | 0 | 0 | 0 | 0 | 0 | 0 | 1 |

**Figure 5.** 3-8 multiplexer model with its truth table. Reproduced with permissions from [35].

Suppose the aforementioned 3-8-line decoder is translated into a non-dispatchable multi-source park. In that case, the developed permutated aggregator at the downstream side of the power transformer has the configuration portrayed in Figure 6. The scheme consists of a 3-8-line decoder, whose feedback is fed by the energy losses gathered at the PCC and is sent to an enabler.

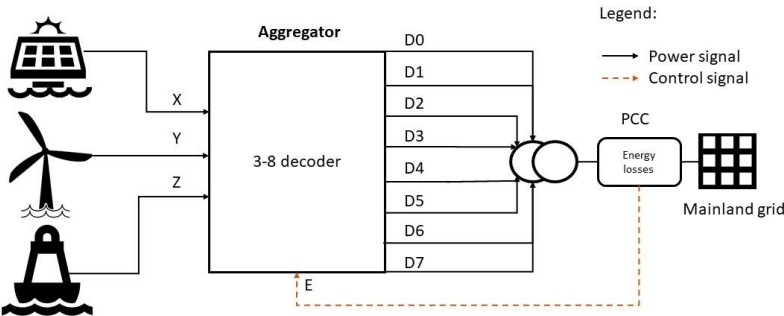

**Figure 6.** 3-8-line-decoder-based energy losses enabled aggregator.

In Table 3, an expanded version of this permutation logic is represented, where the multiplying factor *perc* is assumed to be 0.5, whereas $P_{\text{PV}_{AC-side}}$, $P_{wind}$ and $P_{wave}$ represent the power output coming from OFPV, wind, and wave parks, respectively, *Output* represents the output at the aggregator. The Decode pin represents the performed commutation, and 0 corresponds to a total disconnection of the offshore power parks. It is important to pinpoint that, even though the multiplying factor is 0.5, each park is assumed to be connected at null, half, and full rated power. The output is then calculated throughout certain time intervals to obtain the required energy delivered and lost at the PCC.

**Table 3.** Proposed permutation logic based on 3-8 decoder.

| Decode Pin | Permutation Logic (Equation-Based) |
|:---:|:---:|
| 0 | $Output = 0$ |
| 1 | $Output = perc \cdot P_{\text{PV}_{AC-side}}$ |
| 2 | $Output = P_{\text{PV}_{AC-side}}$ |
| 3 | $Output = perc \cdot P_{wind}$ |
| 4 | $Output = P_{wind}$ |
| 5 | $Output = perc \cdot P_{wave}$ |
| 6 | $Output = P_{wave}$ |
| 7 | $Output = perc \cdot \left( P_{\text{PV}_{AC-side}} + P_{wind} \right)$ |
| 8 | $Output = perc \cdot \left( P_{\text{PV}_{AC-side}} \right) + P_{wind}$ |
| 9 | $Output = P_{\text{PV}_{AC-side}} + perc \cdot \left( P_{wind} \right)$ |
| 10 | $Output = P_{\text{PV}_{AC-side}} + P_{wind}$ |
| 11 | $Output = perc \cdot \left( P_{\text{PV}_{AC-side}} + P_{wave} \right)$ |
| 12 | $Output = perc \cdot \left( P_{\text{PV}_{AC-side}} \right) + P_{wave}$ |
| 13 | $Output = P_{\text{PV}_{AC-side}} + perc \cdot \left( P_{wave} \right)$ |
| 14 | $Output = P_{\text{PV}_{AC-side}} + P_{wave}$ |
| 15 | $Output = perc \cdot \left( P_{wind} + P_{wave} \right)$ |
| 16 | $Output = perc \cdot \left( P_{wind} \right) + P_{wave}$ |
| 17 | $Output = P_{wind} + perc \cdot \left( P_{wave} \right)$ |
| 18 | $Output = P_{wind(kW)} + P_{wave}$ |
| 19 | $Output = perc \cdot \left( P_{\text{PV}_{AC-side}} + P_{wind} + P_{wave} \right)$ |
| 20 | $Output = perc \cdot \left( P_{\text{PV}_{AC-side}} + P_{wind} \right) + P_{wave}$ |
| 21 | $Output = perc \cdot \left( P_{\text{PV}_{AC-side}} \right) + P_{wind} + P_{wave}$ |
| 22 | $Output = P_{\text{PV}_{AC-side}} + perc \cdot \left( P_{wind} + P_{wave} \right)$ |
| 23 | $Output = P_{\text{PV}(kW)_{AC-side}} + P_{wind(kW)} + perc \cdot \left( P_{wave(kW)} \right)$ |
| 24 | $Output = P_{\text{PV}(kW)_{AC-side}} + perc \cdot \left( P_{wind(kW)} \right) + P_{wave(kW)}$ |
| 25 | $Output = perc \cdot \left( P_{\text{PV}(kW)_{AC-side}} \right) + P_{wind(kW)} + perc \cdot \left( P_{wave(kW)} \right)$ |
| 26 | $Output = P_{\text{PV}(kW)_{AC-side}} + P_{wind(kW)} + P_{wave(kW)}$ |

The reason behind the expansion of the proposed aggregator to a broader number of outputs is that, when the three sources theoretically work all at rated power, they sum up to 18-MW, showing that utilizing gates for going the closest to the performance of 3-8 PLC Decoder would not be possible. Implementing such a scheme based on power electronic gates would imply installing an enormous amount of such logic units, which is technically and economically unfeasible [36]. Hence, it is necessary to replicate the same gate logic with genetic programming without using gates [34], where several input/output configurations can be addressed [24].

When all the power sources are disconnected, the Y-connected three-phase load is directly fed by a 100 MW generator coming from the mainland, represented by a transfer function. The permutation logic depicted above is developed and simulated on MAT-LAB/Simulink, using both Simulink and STATEFLOW domains, where only active power components are studied and later subjected to calculations for a 5-year time interval to obtain the corresponding energy outputs.

### 2.4.2. Capacity Factor Calculation Based on the Maximum Energy Output

Electricity flexibility services have historically been provided by a nurtured bunch of power sources, both fossil and non-fossil fuels, with coal and natural gas combined prevailing as the cornerstones and the demand response ranging from 15 to 30% by energy scenario. However, it is expected in the STEPS that $CO_2$-neutral energy technologies, independently of the economy market, surpass 40% of the electricity flexibility mix. Moreover, at least 10% of this flexibility mix is attained from other renewable sources different from hydropower [1], which enables us to envision the potential of variable renewable sources as flexibility service providers without depending on 100% of any storage system [23].

Nonetheless, their ability to provide flexibility services depends on several aspects: rated power, capacity factor, control strategy, and combination or separation of power parks. The capacity factor of a generation system ($Cf$) is obtained by calculating the relationship between the accumulated energy output ($E_{out}$) versus the rated power ($P_{nom}$) of a machine during a specific time interval ($T_{int}$), as depicted in (3) [20]. OFPV, wind, and wave, when constituting separate power parks, can report ranged capacity factors of 10–25%, 25–60%, and 28–41%, respectively, whereas the capacity factors of OFPV–wave and OFPV–wind–wave combined have not yet been obtained as far as the authors are concerned [20,22].

A capacity factor for OPWW power parks at the aggregator is introduced, which entails that Equation (3) is modified to (4):

$$Cf = \frac{E_{out}}{P_{nom} \cdot T_{int}} \qquad (3)$$

$$Cf_{Source1, \, Source \, 2,...,SourceN} = \frac{\sum E_{out(Source1, \, Source \, 2,...,SourceN)}}{\sum P_{rated(Source1, \, Source \, 2,...,SourceN)} \cdot T_{int}} \qquad (4)$$

where $Cf_{Source1, \, Source \, 2,...,SourceN}$ is the capacity factor of sources combined at the aggregator side ($Source1, \, Source \, 2, \ldots, SourceN$), $E_{out(Source1, \, Source \, 2,...,SourceN)}$ is the energy output of these sources at the aggregator, and $P_{rated(Source1, \, Source \, 2,...,SourceN)}$ is the rated power (18 MW for the studied OPWW power parks).

### 2.4.3. Seasonal Permutation Logic with and without Employment of a GA-Based Optimization Technique

A seasonal comparison of all the key performance indicators (KPIs) of interest for this research is carried out (energy losses, capacity factor, individual contribution, etc.). The simulations of permutation logic depicted in the table above are collected for the time intervals 1–31 January, 1–30 April, 1–31 July, and 1–31 October of 2016. Then, an optimization process is carried out, assuming that the permutation logic would not consider a total disconnection from the aggregator. Instead, all three sources are kept connected

to the aggregator, but sometimes some or all of them could contribute zero energy to the cluster.

GA is used to ascertain the optimal combination of the available energy source for minimizing energy losses. The method is a heuristic search procedure based on the logic of the natural selection process where the fittest survive and reproduce for the next generations. The process entails curtailing the available resources (PV, wind, and wave power) in random order when the energy generation exceeds the demand. The excess energy can then be channeled to other loads through, for example, transformer tap changers or onshore metal-clad cells. The power that cannot be deviated to other loads can be redirected to correctly sized storage systems.

Hence, the single-objective function is given by:

$$Obj(min) = E_{losses\text{OPWW}} \tag{5}$$

Subjected to:

$$
\begin{aligned}
PV_{aval} &\geq PV_{dev} \geq 0.3 \cdot PV_{aval} \\
Wind_{aval} &\geq Wind_{dev} \geq 0.3 \cdot Wind_{aval} \\
Wave_{aval} &\geq Wave_{dev} \geq 0.3 \cdot Wave_{aval} \\
if\ Demand &< Generation,\ then\ No\ curtailment \\
if\ Demand &> Generation,\ then\ Curtail \\
if\ after\ curtailment,\ &Demand < Generation,\ then\ No\ curtailment
\end{aligned}
\tag{6}
$$

where $E_{losses\text{OPWW}}$ is the energy losses accounted for the three sources at the PCC, the subscript *aval* stands for available resource, and *dev* stands for the delivered resource. A total of ten generations are performed for each season, with 700 unique combinations analyzed per generation. The algorithm uses a mutation rate of 0.2, and a single-point crossover is used to generate offspring for subsequent generations. The best combination is carried out over to the next generations without mutation. Figure 7 shows the optimization search process chart.

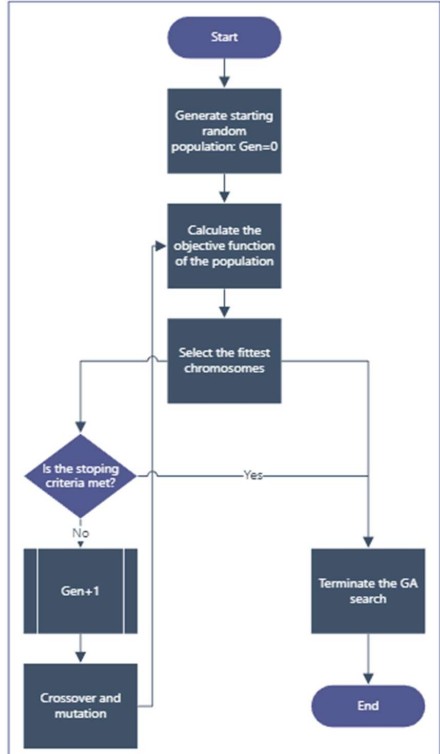

**Figure 7.** GA flowchart.

2.4.4. BESS Sizing after Optimization

For the sake of the performance of the optimization process, the consumption pattern is considered as the energy set-point that each generation system ought to track individually. Once the GA search is terminated, the difference between the total energy generated by the combined power parks and the energy before and after the optimization will be accounted for to undertake BESS sizing.

The steps to size the BESS are described as follows:

(a)  Calculate the energy supplied by the three sources combined based on the energy profiles depicted in Figure 2.
(b)  Start the GA optimization process.
(c)  Calculate the seasonal energy supply after performing the GA search (energy after curtailment).
(d)  Calculate the peak energy curtailed on each season and obtain the average value.
(e)  Calculate the current under the average peak energy obtained per season and the rated power of the main transformer.
(f)  Calculate the BESS size according to the current.
(g)  Calculate the number of batteries needed based on the ampere-hours of each battery.
(h)  Form strings to comply with the voltage and current requirements at the transformer side. If needed, place a dedicated distribution transformer for the final BESS.

Before performing the BESS sizing calculations, several assumptions are made:

(a)  The specifications of each battery are assumed to be 48 V/170 Ah/5 kW, in accordance to design criteria found for commercial Li-ion batteries, such as SIRIUS SuperCap 3550-48-B-1.7C-M-SD-A-G (KiloWatts Labs), VTLF48V-A267 (Vottery), and LPBA48170 (Felicity Solar).
(b)  The required duration or the time during the load that must be supplied is 10 h. Moreover, the BESS can be prepared for peak shaving [37].
(c)  The state of discharge (SoD) is assumed to be 20%.
(d)  No capacity rating or charge/discharge curve is considered.
(e)  The BESS sizing is not included in the GA search. It is calculated afterward.
(f)  No inverter is studied for the BESS. However, it is considered that an arrangement of this nature needs a dedicated distribution transformer if connected to the downstream side of the main transformer (6.6 kV). The rated power of the distribution transformer is estimated according to the peak energy after curtailment.
(g)  The charge/discharge current is set as 105 A (5 kW/48 V).
(h)  No costs are estimated, neither for the batteries nor the distribution power transformer.
(i)  The size of the BESS is calculated with the following:

$$Size_{battLi-ion} = \frac{100 \cdot I \cdot t}{100 - Q} \qquad (7)$$

where $Size_{battLi-ion}$ is the size for Li-ion BESS in A-h, $I$ is the current in amperes, $t$ is the duration in hours, and $Q$ is the remaining charge of the BESS in %, which cannot be under 20% for Li-ion batteries [38].

(j)  After calculating the BESS size, the number of batteries for that is given by the ratio between the BESS size in A-h and the capacity of each battery (170 A-h).
(k)  The transmission and distribution cable arrangements connected to the main transformer are not upgraded.
(l)  Even though the BESS is sized with the help of calculations, there were no simulations undertaken with the arrangements that have been obtained.

Based on the calculations, different BESSs based on power/capacity at the main transformer are proposed.

## 3. Results

### 3.1. Permutation Logic Analysis through Aggregation of Multiple Offshore Renewable Sources

Having performed the permutation logic described in Table 3 for a 5-year hourly dataset gathered from an emplacement in San Francisco Bay Area from 1 January 2016 to 31 December 2020, some KPIs are plotted in Figure 8.

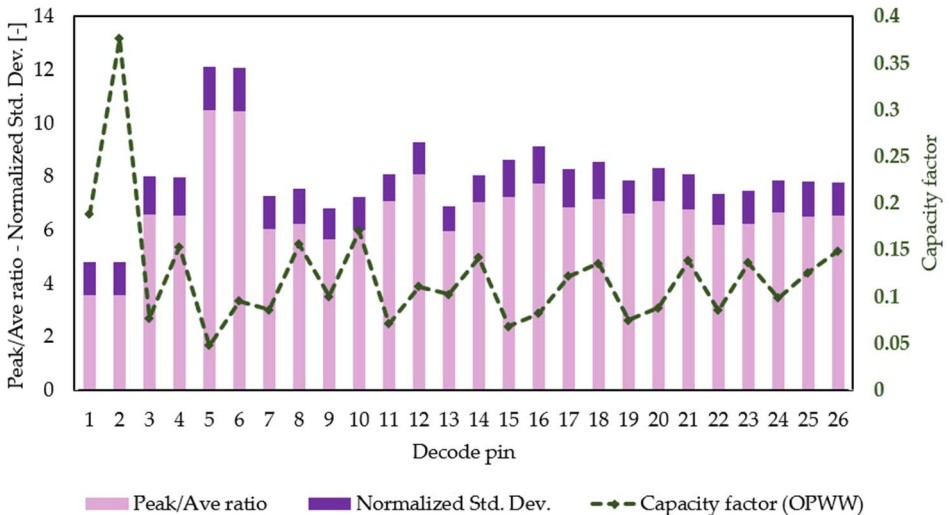

**Figure 8.** Results obtained after performing the permutation logic detailed in Table 3.

Based on the results pictured from now onwards, it can be stated that when either wind or wave energy is curtailed, and their emplacements are connected to an OFPV power park (Decode pins #9 and 13), the peak/average ratio is drastically reduced, which means that an OFPV power park can help to reduce the variability of the energy output to a larger extent. The opposite occurs when wave farms are connected to either OFPV or wind power parks that have been previously curtailed (Decode pins #12 and 16).

Nevertheless, the variability of wave emplacements when these are a part of an individual power park can be beneficial with an important shrinkage of their peak/average ratio when clustered to another power park with different primary sources. Conversely, combining three different sources signifies that such a ratio remains between certain margins, which can be an important indicator of how much the OFPV power park needs installed capacity to reduce variability.

On the other hand, combining two offshore power parks at the same transformer leads to a considerable reduction in the normalized standard deviation. At the same time, tripartite combinations contribute to the opposite. When the combination is wind and wave, the normalized standard deviation remains nearly constant and is even higher compared to triple aggregation systems.

Employing the whole OFPV energy while curtailing either wind or wave energy at the aggregator ameliorates the normalized standard deviation. The combination of OFPV and wave power parks where the wave power is curtailed (Decode pin #13) is the one that provides the shortest value. On the contrary, combining wind and wave power parks while wave energy is curtailed provides the highest magnitude (Decode pin #17), which is a clear indicator that aggregating two power parks instead of three is more beneficial in terms of normalized standard deviation. Thus, the best combination consists of aggregating OFPV and wave power parks, where the PV source is not curtailed (Decode pins #13 and 14).

The capacity factor of combined power parks increases when the power of the PV farm is curtailed. It decreases when the wind power is curtailed and is not substantially affected when the wave power is diminished. Logically, the capacity factor is higher when no source at the power park is curtailed. When the power parks are bipartite, the best combination is OFPV and wind power parks, while the worst is wind and wave power parks.

The results explained above show the benefits of having an OFPV power park as a backup generation system when dealing with overcurrent and overload. It is because the system can handle partial curtailments without relying totally on wind turbines since those, once they are pitched or braked for curtailment purposes, can get worn out at a faster pace, making their maintenance costs higher.

The proposed permutation logic also allows determining how this affects the energy losses at the PCC. As can be observed in Figure 9, several conclusions can be obtained if a capacity factor larger than 0.1 is considered and individual power parks are excluded from the analysis:

(A) For bipartite power parks, the best combination is OFPV and wave power parks (Decode pins #11–14). It diminishes the energy losses and gets a good balance between capacity factor and losses.

(B) In tripartite power parks (OPWW), there is a direct relationship between energy losses and the power park capacity factor. Namely, the greater the power park capacity factor, the higher the losses at the PCC. However, this direct correlation is not manifested in bipartite power parks.

(C) For tripartite power parks, the best balance between energy losses and capacity factor is offered by the combination where wind energy is the only one not curtailed (Decode pin #25). In contrast, curtailing the wind power entails a drastic reduction of both magnitudes (Decode pin #24).

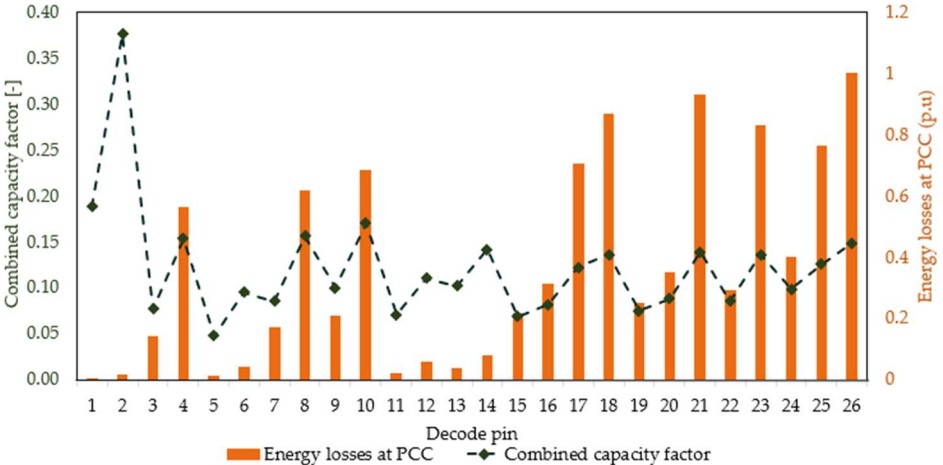

**Figure 9.** Energy losses versus capacity factor.

It is essential to point out that the energy losses portrayed in Figure 9 are calculated per unit using a dividing factor of 2 MWh.

### 3.2. Seasonal Permutation Logic before Optimization

After undertaking the decoded permutation logic described in Sections 2.4.1 and 2.4.2, a seasonal computation of energy losses and capacity factors was performed to compare how well the different proposed combinations work when the PV energy output varies over the yearly time interval under study. Figure 10a represents the energy output after the aggregator, Figure 10b the energy losses at the PCC, and Figure 11 portrays the contribution of each renewable energy (RREE) source to the combined capacity factor, both on a seasonal basis.

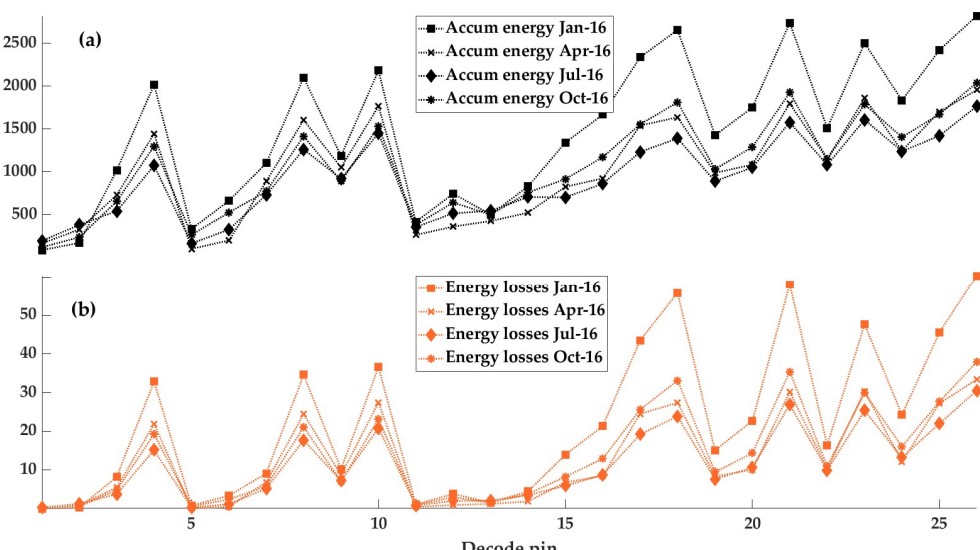

**Figure 10.** Accumulated energy and losses in 2016 (MWh)—seasonal comparison. (**a**) Accumulated energy for 4 months representing the seasons of 2016; (**b**) Energy losses for 4 months representing the seasons of 2016.

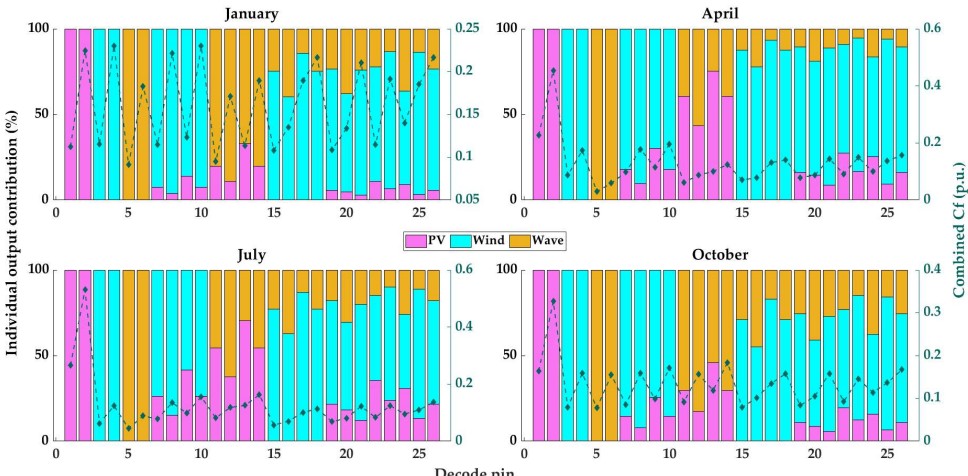

**Figure 11.** Seasonal comparison of the contribution of each source to the capacity factor.

It can be observed that OFPV energy contributes substantially to reducing energy losses during the summer in comparison to winter. Although the wind turbine contributes the most to the total energy poured out to the aggregator, it also contributes to a higher combined capacity factor despite its higher variability, mainly due to its larger installed capacity. Further details about these results will be discussed in Section 4.

It is important to point out that, as stated in Section 2.1.3, the demand is assumed as a resistive three-phase load where the demand curve is only used as a multiplying factor related to the generation profile. Hence, the demand and generation are shifted, and the demand is always covered by offshore energy sources.

### 3.3. Seasonal Permutation Logic with Employment of a GA-Based Optimization Technique

According to the results displayed in Figures 12 and 13, the following can be observed.

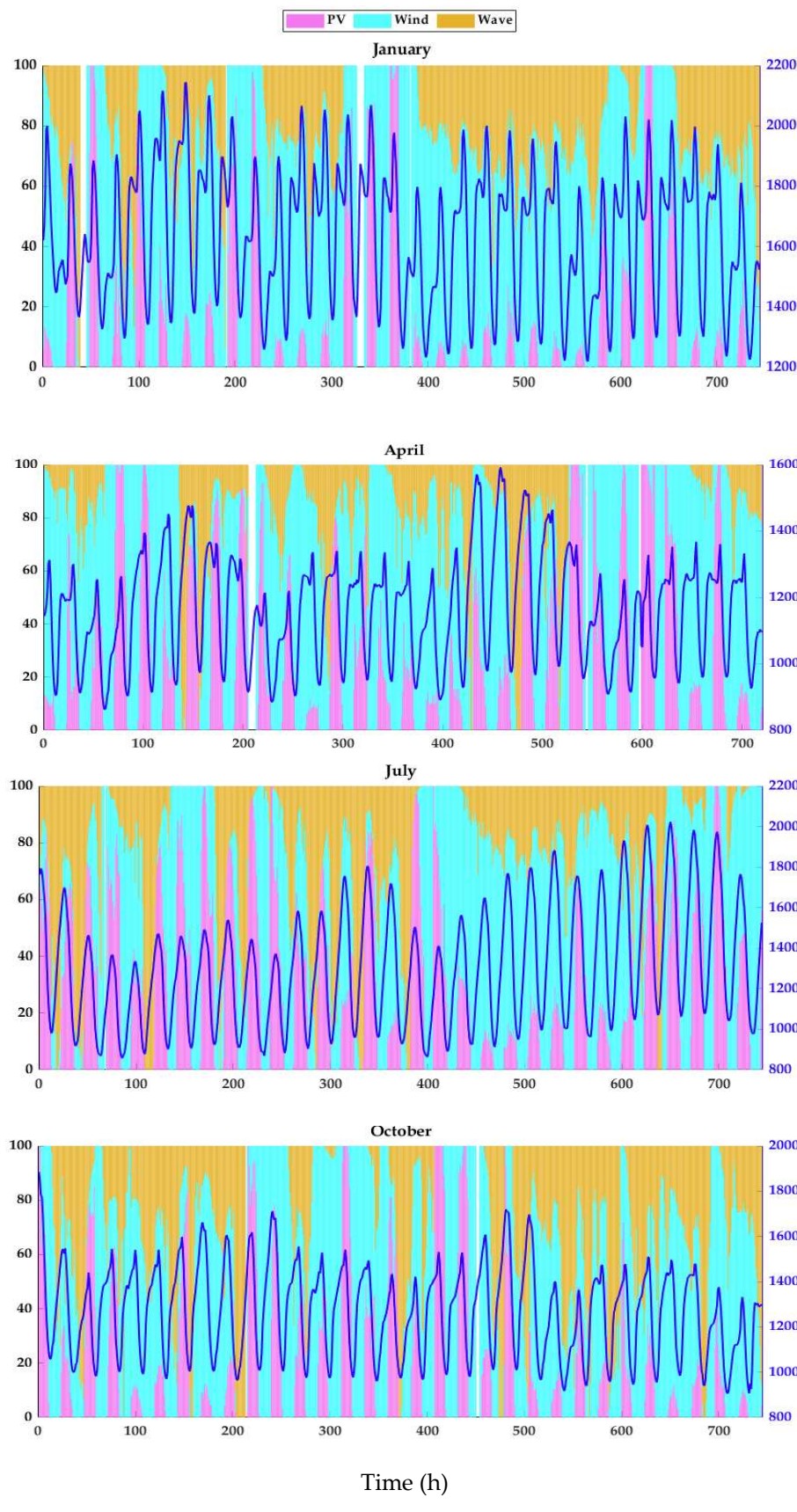

**Figure 12.** Individual contribution vs. demand—seasonal comparison after optimization.

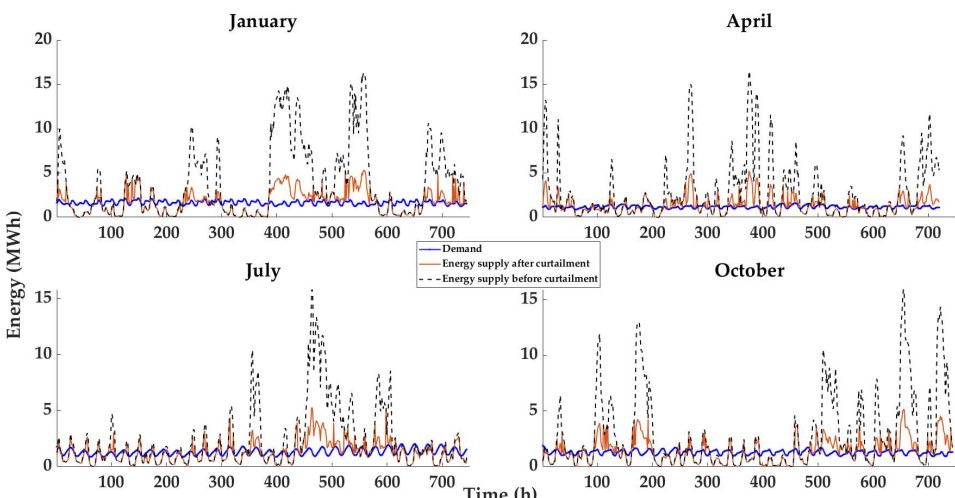

**Figure 13.** Demand vs. supply before and after optimization—seasonal comparison.

(a)　In some instances, not all sources may be present; hence, the available sources are used to fulfill the demand.

(b)　When one or more sources are unavailable, the GA combines what is available and sets the other(s) to zero.

(c)　When none of the sources are available, the energy supply is set to zero, and the algorithm continues to the next day (notice the white bars on the plots in Figure 12).

Further, it can be noticed that either OFPV or wave alone can handle more than 50% of their generation to track the consumption pattern in some instances. It is also observed in Figure 13 that it is not always possible to curtail all the generation to match it entirely with the demand, which could imply that the energy supply surplus after curtailment (red line) must be sent to a correctly sized energy storage system. In contrast, the rest of the energy (before curtailment, the area between dotted black and solid red lines) could be deviated through metal-clad cells towards another population settlement in the same county.

Another important part of the proposed optimization technique is the combined capacity factor. When the optimization is not yet performed, the capacity factor varies between 12 and 20% depending on the season, whereas that variability is considerably reduced after calculating the single-objective function, as can be observed in Table 4. Nonetheless, the capacity factor is also drastically reduced despite its lower variability because the algorithm prioritizes the source that varies the most on a seasonal basis (OPFV) [22]. Two possible solutions for this problem would be either maximizing the combined capacity factor or minimizing the wind turbine's contribution, along with minimizing power losses, which is out of the scope of this research.

**Table 4.** Combined capacity factor before and after seasonal optimization.

| $Cf_{OPWW}$ (%) | 16 January | 16 April | 16 July | 16 October | Standard Deviation |
|---|---|---|---|---|---|
| Before optimization | 20.69 | 14.34 | 12.99 | 15.44 | 3.37 |
| After optimization | 9.69 | 7.84 | 7.27 | 7.74 | 1.07 |

Last but not least, it is encountered that energy losses dramatically decrease after performing the single-objective optimization proposed in this paper, being smaller during the summer and larger during the spring, as indicated in Figure 14. It may be because the demand is lower during the spring and higher during the summer. Hence, the lower the demand, the more sources have to be curtailed. Still, it is important to note that the energy losses before curtailment are higher because the demand is considered inferior compared to the available renewable energy at the aggregator.

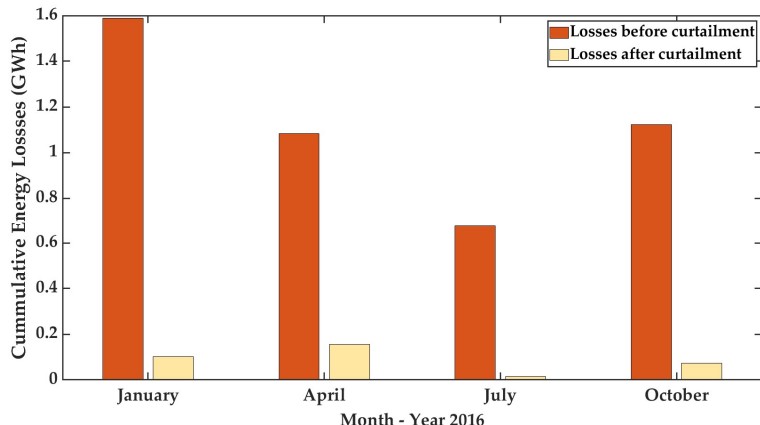

**Figure 14.** Energy losses per season before and after curtailment.

*3.4. BESS Sizing after Optimization*

The difference between the energy before and after curtailment was calculated once the GA search per season was terminated. After, the average energy of 13,000 kWh based on the peak values was obtained.

Considering the previous value and assuming that a dedicated distribution power transformer might be needed, it was necessary to check again the transformer database provided by DTOceanPlus [32]. It was found that a 10-MW 0.69/6.6 kV power transformer can be suitable for connecting the whole BESS offshore while preserving the charging factor of the main transformer below 75%.

Moreover, three different BESS with different series-parallel connections have been obtained, whose main results are presented in Table 5. From the table below, it can be observed that placing a BESS onshore avoids the need to utilize a distribution power transformer. Nevertheless, due to higher voltage ratings, the amount of batteries needed surpasses five times the required number of batteries at the downstream side of the main transformer. Hence, a cost estimation considering onshore/offshore space requirements, transformer sizing, depth of discharge, capacity rating, cable upgrading, and location constraints is recommended.

**Table 5.** BESS sizing with different string arrangements.

|  | Case 1 | Case 2 | Case 3 |
|---|---|---|---|
| Battery rating (single unit) |  | 5 kW, 48 V |  |
| Main transformer rating |  | 40 MW, 6.6/66 kV |  |
| Dedicated BESS transformer rating | 10 MW, 0.69/6.6 kV | 10 MW, 0.69/6.6 kV | N/A |
| Connection side | 0.69 kV | 6.6 kV | 66 kV |
| Location | offshore | offshore | onshore |
| Number of batteries in series | 18 | 14 | 1375 |
| Number of rows | 64 | 85 | 4 |
| Total number of batteries | 1152 | 1190 | 5500 |

## 4. Discussion

*Seasonal Permutation Logic without Optimization*

In this section, results related to seasonal comparisons of the different permutations carried out are discussed. After performing calculations, it was evidenced that when yearly datasets are compared, differences are barely perceptible, if not inexistent, and that is the reason why such results are not shown. However, for an entire year, e.g., 2016, important deviations are encountered, as depicted in Figures 10–14. Several insights can be obtained from these:

(A) For individual power parks, the higher the rated power, the higher the energy losses, which is expected considering that the studied PV emplacement has scales of 1:5 and 1:12 if compared to wave and wind power parks, respectively.

(B) When the power parks are bipartite, it is evidenced that curtailing wind power is more beneficial when it is connected to either PV or wave emplacements, but when PV and wave emplacements share the cluster, curtailing wave energy is a better solution when no storage system is available. Further, combining OFPV and wave reports additional benefits at this particular emplacement: a more drastic reduction of energy losses. However, these do not overmuch deviate regardless of the season. Due to wind speed being inversely proportional to temperature, wave energy can play the role of a " storage system" when OFPV emplacement is no longer available because of a lack of irradiance.

(C) Curtailing OFPV energy from combined OFPV and wind power parks does not significantly decrease losses. However, these are considerably lower during summer–fall (July or October) than during winter–spring (January or April). That proves that increasing the installed capacity of PV emplacements is the best way to go due to the capacity of wind turbines increasing, but the rated power ratio between PV and wind emplacements shall be high enough to reach a balance between energy losses and seasonal variability.

(D) The power park capacity factor, although highly variable depending on the sources combined at the aggregator, is higher during the winter and lower during the summer, except in those cases where there is a bipartite OFPV and wave power park. In such a case, the capacity factor is reduced during the spring.

(E) Between spring and fall, the capacity factor slightly deviates, seasonally speaking, independently of the combination, whereas, between winter and summer, the deviation is no longer slight. On top of that, curtailing OFPV over wind or wave energy on bipartite combinations substantially increases the capacity factor, especially in those seasons where irradiance impacting the PV panels is lower. It can open the gate for adjusting the rated power ratio instead of using large energy storage systems that would become economically unfeasible if placed on dedicated offshore floaters.

(F) On tripartite power parks, the capacity factor can be higher whether either OFPV or wave energy is curtailed or not, whereas it is drastically reduced when wind energy is curtailed. Indeed, curtailing only OFPV energy barely affects the performance of the power park in terms of seasonal capacity factor and energy losses but can drastically deprecate the energy losses when measured for a wider time interval, as depicted in Figure 9 (Decode pins #21 and 26). The same applies when wave energy is curtailed over wind and/or OFPV energy (Decode pins #23 and 25).

(G) If we look at Figures 10 and 11 together, with special emphasis on Decode pins ranging from #19 to #26 (see Table 3), it is evidenced that the energy losses attributed to the combination of the three power sources become larger with less curtailment in renewable sources. Additionally, the capacity factor varies considerably among seasons, which is not a desirable behavior of the system.

(H) Once the optimization process is performed, it is evidenced that the main goal of the proposed single-objective function is achieved, and the energy losses are strongly dependent on seasons. Further, it is not always possible to curtail all the generation sources to match the consumption pattern. Hence, it is strongly advisable to include storage systems that are properly sized and complement them with metal-clad cells on substations to deviate the energy curtailed to another population settlement in the same county, which can be completed in the mainland substation and is cheaper.

(I) Despite the variability reduction in the combined capacity factor, it is evidenced that this variable has to be included as a variable in a multi-objective optimization process to obtain a higher magnitude.

(J) Since offshore wind turbines are currently being designed for higher-rated power, it is advisable to increment in a reasonable proportion the installed capacity of OFPV, even

though the combined capacity factor can be lowered. This is particularly important for those sites where the peak demand is almost coincident with the peak generation, such as the site studied for this research.

(K) Lastly, when the demand curve is directly used as the set-point for generation, it is evidenced that each power source accommodates itself in such a way that it can track individually the consumption pattern, which unveils the potential of renewable energies as flexibility service providers even in the absence of storage systems [1].

(L) If the BESS is placed onshore, it will require a considerable number of battery units, but they might not need to include a distribution power transformer offshore, which avoids the necessity of building floaters or foundations, as well as upgrading the main transformer and cables, which, consequently, reduces the costs.

(M) It is recommended to perform BESS sizing during the GA search and compare it to the battery sizing performed afterward. Thus, the optimization process should include this sizing as another variable subjected mainly to space, location, and cost constraints.

## 5. Conclusions

A novel GA-based permutation control logic has been applied to an aggregator for a pure offshore multi-source park. The aggregator has been designed under the principles of genetic programming and mimics the behavior of an electronic 3-8 multiplexer. The optimization technique was tested for a power park with input data measured at a location in the San Francisco Bay Area. The main conclusions that can be derived from this study are:

(a) The proposed permutated aggregator fulfills its primary function, allowing the power parks to contribute partially and individually to diminish the energy losses at the PCC, which eliminates the need to disconnect any source.

(b) Genetic programming and GA are a good match when it comes to performing permutated control on multi-source parks, which can help to improve the performance of the transformers' on-load tap changers (OLTC). However, more research on this topic is needed.

(c) The capacity factor of the multi-source park is improved in terms of seasonal variability (standard deviation), although its value is considerably reduced when the demand is utilized as a set-point to be individually tracked.

(d) Without any storage system involved, the multi-source park has demonstrated to be capable of providing flexibility services towards mainland grids, which is aligned with the new energy policies stated in WEO 2021.

(e) A reasonable proportion between OFPV/wind/wave power parks is advisable due to the considerably higher capacity of the wind turbines. Thus, the proposed GA-based permutation logic would not rely too much on the partial braking of wind turbines, which can be detrimental to their performance.

(f) Even though the studied energy technologies are capable of providing individual flexibility services, it is not always possible to curtail generation to track the demand at the same pace. Hence, optimized storage sizing is recommendable.

The future scopes of this research include the addition of optimized storage sizing, combined capacity factor, penalty costs, and rated power of the wind turbine as decision variables of a multi-objective function, subjected to several constraints, such as power curtailment, cloud shading, hydrodynamic (wave power), and aerodynamic forces (wind power).

**Author Contributions:** Conceptualization, B.R.-D.; methodology, B.R.-D. and C.E.; software, B.R.-D. and C.E.; validation, B.R.-D., C.E., G.L. and I.T.; formal analysis, B.R.-D. and C.E.; investigation, B.R.-D., C.E. and G.L.; resources, B.R.-D. and G.L.; data curation, B.R.-D., C.E. and G.L.; writing— original draft preparation, B.R.-D. and C.E.; writing—review and editing, G.L. and I.T.; visualization, B.R.-D. and C.E.; supervision, I.T.; project administration, I.T.; funding acquisition, I.T. All authors have read and agreed to the published version of the manuscript.

**Funding:** This work is supported by the EU-SCORES project financed by the European Union's Horizon 2020 and Green Deal Research and Innovation Programme under grant agreement No 101036457, STandUP for Energy and Uppsala University.

**Data Availability Statement:** The data presented in this study are available on request from the corresponding author. The data are not publicly available due to privacy restrictions.

**Conflicts of Interest:** The authors declare no conflict of interest.

## Nomenclature

| | |
|---|---|
| AC | Alternate Current |
| BESS | Battery energy storage system |
| DC | Direct current |
| DER | Distributed Energy Resources |
| DSM | Demand-Side Management |
| GA | Genetic Algorithm |
| KPIs | Key Performance Indicators |
| MPPT | Maximum Power Point Tracking |
| OC | Open-circuit |
| OFPV | Offshore floating Photovoltaic Power |
| OPWW | Offshore Photovoltaics, Wind, and Wave Power |
| PCC | Point of Common Coupling |
| RES | Renewable Energy Sources |
| SC | Short-circuit |
| STEPS | Stated Policies Scenario |
| WEC | Wave energy converter |
| WEO | World Energy Outlook |

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
