# Peer review of "GA-Based Permutation Logic for Grid Integration of Offshore Multi-Source Renewable Parks"

_machines, doi:10.3390/machines10121208_

Round 1

Reviewer 1 Report

In the article, a GA-based control for various types of energy source coupling was proposed and analyzed. 

The bibliographic research was good enough. The use of real data e well-described equipment is a strong point. Each aspect of the work was discussed and clearly present the conclusions and highlights.

My only concern is about the diversity of the graphics designs, perhaps some of them could be uniformized.  Figure 11 could be miniaturized to stand along the text and not in a single page.

Reviewer 2 Report

The paper proposes a Genetic Algorithm based permutation control logic applied to the aggregator of an offshore multi-source park. The energy losses at common coupling point is accounted for the feedback. This paper focuses on distributed energy resources such as Floating PV, wind power and wave offshore power sources.

It woul be an interesting add to show the storage sizing and its comparisons to the current case study. Also to see the performance under different power/capacity.

I strongly suggest the authors to revisit some figures such as Figure 11, to make sure the indentation and legibility is better and easier for the readers.
